# PolySketchFormer: Fast Transformers via Sketches for Polynomial Kernels

## Abstract

The quadratic complexity of attention in transformer architectures remains a big bottleneck in scaling up large foundation models for long context. In fact, recent theoretical results show the hardness of approximating the output of softmax attention mechanism in sub-quadratic time assuming Strong Exponential Time Hypothesis. In this paper, we show how to break this theoretical barrier by replacing softmax with a polynomial function and polynomial sketching. In particular we show that sketches for Polynomial Kernel from the randomized numerical linear algebra literature can be used to approximate the polynomial attention which leads to a significantly faster attention mechanism without assuming any sparse structure for the attention matrix that has been done in many previous works.

In addition, we propose an efficient block-based algorithm that lets us apply the causal mask to the attention matrix without explicitly realizing the $n \times n$ attention matrix and compute the output of the polynomial attention mechanism in time linear in the context length. The block-based algorithm gives significant speedups over the *cumulative sum* algorithm used by Performer to apply the causal mask to the attention matrix. These observations help us design *PolySketchFormer*, a practical linear-time transformer architecture for language modeling with provable guarantees.

We validate our design empirically by training language models with long context lengths. We first show that the eval perplexities of our models are comparable to that of models trained with softmax attention. We then show that for large context lengths our training times are significantly faster than FlashAttention.

## 1 Introduction

Transformer (Vaswani et al., 2017) based models are state-of-the-art for many Natural Language tasks and led to breakthroughs in tasks such as machine translation, language understanding (Devlin et al., 2019) and language modeling (Brown et al., 2020; Chowdhery et al., 2022; OpenAI, 2023; Anil et al., 2023). Due to the quadratic time complexity of the attention mechanism, transformer based models have been limited to short context lengths. Numerous variants of the vanilla transformer have been proposed to address the quadratic time complexity (Wang et al. (2020); Katharopoulos et al. (2020); Choromanski et al. (2020) and many more). These variants are colloquially referred to as "Efficient Transformers". A recent survey by Tay et al. (2022) provides a broad overview of different techniques that have been employed to obtain approximations[1] of the attention mechanism. While many of the efficient transformer constructions have a theoretical per step training latency that is linear in the context length, the survey observes that the training latency improvements provided by many of these constructions in practice have been disappointing and also at a loss in model quality. They also note that most state-of-the-art models still use the vanilla transformer. In this work, we focus on improving the training latency of transformer models in the decoding-only tasks such as language modeling trained via next word prediction objective. Our techniques generalize to the encoding-only and encoder-decoder transformers as well.

Another line of work (Dao et al., 2022; Dao, 2023), termed as FlashAttention and FlashAttention-2, towards enabling the training of transformers on large context lengths, looked at more efficient

---

[1]Here, the word approximation is to be treated informally. Many "efficient transformer" constructions deviate significantly from the vanilla transformer.

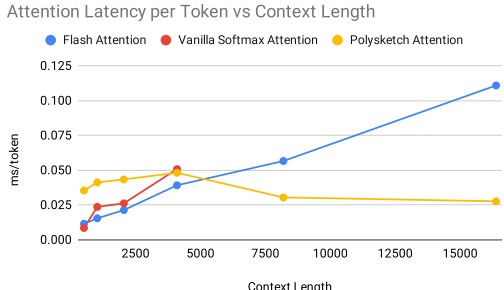

Figure 1: The plot compares attention latency per token during training of a model at different context lengths (512, 1k, 2k, 4k, 8k, 16k). We can see that the while the attention latency of PolySketch-Former remains in a small window irrespective of the context length, the attention latencies of vanilla softmax attention and flash attention grow linearly with the context length. Our model with vanilla softmax attention runs out of memory for context lengths greater than 4k.

implementations of the standard attention mechanism. Using techniques such as blocking and rematerialization, they showed that vanilla attention mechanism can be implemented without realizing the full $n \times n$ (here $n$ denotes the context length, i.e., the number of input tokens) attention matrix in the High-Bandwidth Memory (HBM) of ML accelerators (such as GPUs/TPUs). Along with other techniques such as operator fusion and more efficient work partitioning, they showed that transformers up to a context length as large as $16k$ can be trained efficiently. The fact that $O(n^2)$ memory is not required to train transformers has also been noted in an earlier work of Rabe & Staats (2021). While FlashAttention and FlashAttention-2 reduce the amount of High-Bandwidth Memory (HBM) required to train a transformer model, the amount of computation per training step still scales as $O(n^2)$ (see Figure 1) and hence scaling to large context lengths is hard even with a very efficient implementation of the vanilla transformer model. Thus, it is imperative to obtain an efficient *linear* transformer[2] that improves upon the training latency of efficient implementations of *vanilla* transformers in practice while not losing on the performance of vanilla transformers.

Recently Hua et al. (2022) proposed a variant of Transformer which uses a Gated Attention Unit instead of the usual Multi-head Softmax Attention + Feed Forward layers. They argue that previously proposed efficient transformers suffer from inferior quality (as compared to suitably augmented vanilla transformers), have overheads in practice and are inefficient during auto-regressive training because of an RNN-style sequential dependence of linear transformers which essentially renders the linear attention implementation to be memory-bandwidth bound. They propose a mixed-chunk based mechanism which at a high-level works as follows: (i) They first chunk the input sequence into segments of length $C$ each, (ii) within each chunk, apply quadratic attention using relu$^2$ instead of normalized softmax to obtain the attention matrix (appropriately masked for decoding-only transformers), and (iii) across chunks, apply a global linear transformer (causal for decoding-only transformers). They then add up the outputs of the local and global attention matrices and then use the result to gate the output of a dense layer that applies a linear transformation to each token. They show that this local + global attention mechanism has comparable performance to an augmented version of Transformer they refer to as Transformer++[3]. While they show that the perplexities of Transformer++ can be matched with their model, it is unclear how successful the "linear global attention" mechanism is in capturing the long range dependence. Since the chunk size they use in their experiments (256) is large enough and they locally use a strong attention mechanism (relu$^2$), the perplexities in language modeling can be extremely good even while failing to capture long range dependencies (Rae & Razavi, 2020). Thus we seek to explore efficient transformers that *uniformly* apply a strong attention mechanism to all the tokens in the context window.

The starting point to our work is the Performer (Choromanski et al., 2020), a kernel based Transformer that provably approximates the softmax attention matrix. The kernel based view of attention matrix was also taken by other earlier works (Tsai et al., 2019; Katharopoulos et al., 2020). We will briefly review the softmax attention mechanism in vanilla transformers and explain how Performer works. Let $\mathbf{X} \in \mathbb{R}^{n \times d}$ be the pre-projection input to the attention mechanism, where each row corresponds to a token in the context. Let $\mathbf{W}_q, \mathbf{W}_k$ and $\mathbf{W}_v \in \mathbb{R}^{d \times h}$ be the weight matrices corresponding to the *query*, *key* and *value* projections. The vanilla attention mechanism defines the attention-weights matrix $\mathbf{A} \in \mathbb{R}^{n \times n}$ as $\mathbf{A} = \text{softmax}(\text{mask}(\mathbf{XW}_q(\mathbf{XW}_k)^\mathsf{T})/\sqrt{h})$ and the output

---

[2]Linear transformers denote models that have a training latency that scales linearly in the context length.

[3]Concretely, Transformer++ refers to a vanilla Transformer + RoPE (Su et al., 2021) as position embeddings before the application of attention block + Gated Linear Units instead of MLPs as the FeedForward layer

of the attention mechanism is defined as $\mathbf{A} \cdot \mathbf{V} \in \mathbb{R}^{n \times h}$, where $\mathbf{V} = \mathbf{X}\mathbf{W}_v$. Here $\mathrm{mask}(\cdot)^4$ denotes a causal mask in the case of decoding-only transformers and the $\mathrm{softmax}(\cdot)$ refers to applying the softmax operator, defined as $[\mathrm{softmax}(x)]_i \doteq \exp(x_i)/\sum_j \exp(x_j)$, on each *row* $x$ of the matrix independently. From now on, we define $\mathbf{Q} \doteq \mathbf{X}\mathbf{W}_q$, $\mathbf{K} \doteq \mathbf{X}\mathbf{W}_k$ and $\mathbf{V} \doteq \mathbf{X}\mathbf{W}_v$ so that the attention weights matrix $\mathbf{A}$ can be succinctly written as $\mathbf{A} = \mathrm{softmax}(\mathrm{mask}(\mathbf{Q}\mathbf{K}^\mathsf{T})/\sqrt{h})$.

Let $\mathbf{q}_i$ and $\mathbf{k}_j$ denote the $i$-th and $j$-th rows of the matrices $\mathbf{Q}$ and $\mathbf{K}$ respectively. By definition, we have that the $(i,j)$-th entry of matrix $\mathbf{A}$ is given by

$$\mathbf{A}_{i,j} = \frac{\mathrm{mask}(i,j)\exp(\langle \mathbf{q}_i, \mathbf{k}_j \rangle/\sqrt{h})}{\sum_{j'} \mathrm{mask}(i,j')\exp(\langle \mathbf{q}_i, \mathbf{k}_{j'} \rangle/\sqrt{h})}.$$

In the training of decoding-only transformers, we have $\mathrm{mask}(i,j) = 1$ if $i \geq j$ and $0$ otherwise. Hence the attention matrix $\mathbf{A}$ is lower-triangular. Choromanski et al. (2020) propose FAVOR+ (Fast Attention Via Positive Orthogonal Random features) mechanism showing that the query key vectors $\mathbf{q}_i$ and $\mathbf{k}_j$ can be mapped using a random non-linear mapping to *nonnegative* vectors $\tilde{\mathbf{q}}_i$ and $\tilde{\mathbf{k}}_j$ each of a dimension $m = \Theta(h \log h)$ such that with *any* constant probability over the random mapping, with $\tilde{\mathbf{A}}_{i,j}$ defined as $\tilde{\mathbf{A}}_{i,j} := \frac{\mathrm{mask}(i,j)\langle \tilde{\mathbf{q}}_i, \tilde{\mathbf{k}}_j \rangle}{\sum_{j'} \mathrm{mask}(i,j')\langle \tilde{\mathbf{q}}_i, \tilde{\mathbf{k}}_{j'} \rangle}$, $\max_{i,j} |\tilde{\mathbf{A}}_{i,j} - \mathbf{A}_{i,j}|$ is small. Using associativity of matrix multiplication, one can show that $\tilde{\mathbf{A}} \cdot \mathbf{V}$ can be computed in time linear in the context length for both causal and non-causal masks. We note the following important qualifications on FA-VOR+ attention results: (i) The result assumes bounded $\ell_2$ norms of the query and key vectors. The dimension $m$ has to grow exponentially in the squared $\ell_2$ norm of the query and key vectors to obtain that $|\tilde{\mathbf{A}}_{i,j} - \mathbf{A}_{i,j}|$ is small. (ii) The result only holds for a given fixed set of query and key vectors. In particular, a single randomized mapping does not preserve the attention matrix for *all* possible inputs even under the bounded norm assumption. Nevertheless, the experiments in Choromanski et al. (2020) show that FAVOR+ works well in practice and show that the attention computation using FAVOR+ scales linearly in the context length when used in the context of non-causal attention.

A major goal of FAVOR+ is to approximate the softmax attention matrix but the restriction to only queries and key vectors with bounded $\ell_2$ norms can be a severe drawback. A recent work of Alman & Song (2023) shows that it is not possible to (entry-wise) approximate the output of the attention mechanism in $o(n^2)$ time assuming Strong Exponential Time Hypothesis (SETH) unless the magnitudes of the entries of the $\mathbf{Q}, \mathbf{K}, \mathbf{V}$ matrices are bounded by $\Theta(\sqrt{\log n})$ for $h = O(\log n)$. While this negative-result does not say that attention approximation is hard when using other forms of error metrics, it does show that there are significant barriers to be overcome to approximate the output of the attention mechanism in $o(n^2)$ time without further assumptions on the query and key vectors.

Motivated by the barriers present in obtaining fast algorithms to approximate the output of softmax attention mechanism, we explore the use of polynomials instead of softmax in the definition of attention mechanism. Concretely, for even degree $p$, we define the attention weights matrix $\mathbf{A}^{(p)}$ as

$$(\mathbf{A}^{(p)})_{i,j} = \frac{\mathrm{mask}(i,j)\langle \mathbf{q}_i, \mathbf{k}_j \rangle^p}{\sum_{j'} \mathrm{mask}(i,j')\langle \mathbf{q}_i, \mathbf{k}_{j'} \rangle^p}.$$

We use an even degree $p$ to ensure that the entries of the attention matrix $\mathbf{A}^{(p)}$ are nonnegative and retain the interpretation of rows of the matrix $\mathbf{A}^{(p)}$ defining probability distributions over the tokens. For a causal mask, which is the focus of our work, the polynomial attention weights matrix $\mathbf{A}^{(p)}$ can be written as

$$\mathbf{A}^{(p)} = \mathbf{D}^{-1} \cdot \mathrm{lt}_\triangle((\mathbf{Q}\mathbf{K}^\mathsf{T})^p) \text{ where } \mathbf{D} = \mathrm{diag}(\mathrm{lt}_\triangle((\mathbf{Q}\mathbf{K}^\mathsf{T})^p) \cdot \mathbf{1}). \tag{1}$$

In the above expressions, for a matrix $\mathbf{M}$ we use $\mathbf{M}^p$ to denote the matrix obtained by powering each entry of $\mathbf{M}$ by degree $p$ and $\mathrm{lt}_\triangle(\mathbf{M})$ denotes the matrix obtained by only keeping the lower-triangular entries of the matrix $\mathbf{M}$ and zeroing the rest of the entries. There is a simple algorithm that can compute $\mathbf{A}^{(p)}$ with only a linear dependence on $n$ for both causal and non-causal masks. For a vector $\mathbf{q} \in \mathbb{R}^d$, let $\mathbf{q}^{\otimes p}$ be the $h^p$ dimensional vector obtained by tensoring $\mathbf{q}$ with itself for $p$ times. Concretely, we index the coordinates of $\mathbf{q}^{\otimes p}$ with a $p$-tuple $(i_1, \ldots, i_p) \in 1, \ldots, h^p$ and define

---

$^4$For any matrix $\mathbf{M}$, $\mathrm{mask}(\mathbf{M})$ indicates an matrix such that the $(i,j)$-th entry is $\mathrm{mask}(i,j) \cdot \mathbf{M}_{i,j}$ where $\mathrm{mask}(i,j) \in \{0,1\}$.

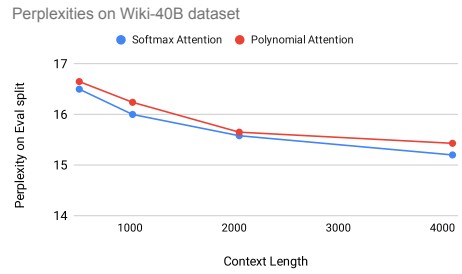 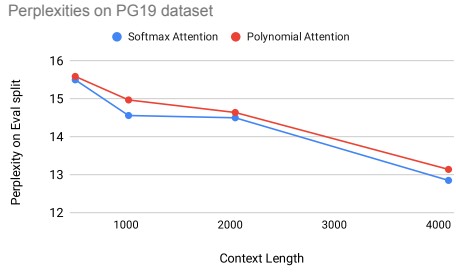

(a) Perplexities on Eval Split of Wiki-40B dataset    (b) Perplexities on Eval Split of PG-19 dataset

Figure 2: Perplexities attained by models trained using softmax attention and degree-4 polynomial attention two datasets: (i) Wiki-40B and (ii) PG-19 at various context lengths (512, 1k, 2k, 4k). At all context lengths, the polynomial attention achieves perplexities close to that of softmax attention.

$(\mathbf{q}^{\otimes p})_{(i_1,\dots,i_p)} \coloneqq \mathbf{q}_{i_1} \cdots \mathbf{q}_{i_p}$. Now, for any two vectors $\mathbf{q}$ and $\mathbf{k}$, we have $\langle \mathbf{q}^{\otimes p}, \mathbf{k}^{\otimes p} \rangle = \langle \mathbf{q}, \mathbf{k} \rangle^p$. Given matrices $\mathbf{Q}$ and $\mathbf{K}$, define $\mathbf{Q}^{\otimes p}$ and $\mathbf{K}^{\otimes p}$ be the matrices obtained by tensoring each of the rows of $\mathbf{Q}$ and $\mathbf{K}$ for $p$ times respectively. We then have $(\mathbf{Q}\mathbf{K}^\mathsf{T})^p = \mathbf{Q}^{\otimes p}(\mathbf{K}^{\otimes p})^\mathsf{T}$. In the non-causal case, we have $\mathbf{D} = \mathrm{diag}((\mathbf{Q}\mathbf{K}^\mathsf{T})^p \cdot \mathbf{1}) = \mathrm{diag}(\mathbf{Q}^{\otimes p}(\mathbf{K}^{\otimes p})^\mathsf{T}\mathbf{1})$. Clearly, we can compute $\mathbf{Q}^{\otimes p}(\mathbf{K}^{\otimes p})^\mathsf{T}\mathbf{1}$ using $O(nh^p)$ operations. Similarly given another $n \times h$ matrix $\mathbf{V}$, we can compute $\mathbf{Q}^{\otimes p}(\mathbf{K}^{\otimes p})^\mathsf{T}\mathbf{V}$ using $O(nh^{p+1})$ operations. Hence, non-causal attention with degree-$p$ polynomial has a time complexity that is linear in the context length.

For the causal case, the cumulative sum algorithm presented in Choromanski et al. (2020) shows that the matrix $\mathbf{A}^{(p)}$ defined in (1) can be computed exactly using $O(nh^{p+1})$ operations without having to explicitly compute the matrix $\mathsf{lt}_\triangle((\mathbf{Q}\mathbf{K}^\mathsf{T})^p)$. Indeed, this does show that the computational complexity barriers present in approximating the softmax attention matrix are not present in polynomial attention and we can have an exact linear time algorithm (although impractical since $h^p$ can be quite large even for $p = 4$ for reasonable values of $h$ such as $64$ or $128$ used in today's largest models).

The theoretical linear time complexity for exact polynomial attention motivates us to explore it further. Polynomial has been previously explored to replace softmax in the context of multi-class classification (De Brebisson & Vincent, 2015; Blanc & Rendle, 2018). Tsai et al. (2019) explore polynomial attention mechanism using a degree-2 polynomial but mostly for the purpose of understanding the attention mechanism and not for efficiency. While works such as Schlag et al. (2021) study the model capacity bounds presented by linear time transformers and show using a synthetic model that the recall of tokens is worse compared to the quadratic softmax attention, it is unclear how much of the limitations transfer to full fledged multi-head attention models that are in use today since the simple argument of the necessity of $n$ orthogonal vectors for perfect recall over $n$ tokens is not valid in the multi-head attention case. Our experiments show (see Figure 2) that the perplexities achieved by polynomial attention with $p = 4$ nearly match that of the softmax attention models. Though, perplexity may not capture all the *good* properties of softmax attention, the experiments do show that polynomial attention is a promising candidate to be used in the attention mechanism.

While the theoretical time complexity of polynomial attention is linear in the context length, a major issue in the linear time algorithm presented above to compute the polynomial attention matrix is the multiplicative dependence on $h^p$. To solve for this issue, we use sketches for the polynomial kernel (Ahle et al., 2020) to approximate the attention-weights matrix. Concretely, given matrices $\mathbf{Q} \in \mathbb{R}^{n \times h}$ and $\mathbf{K} \in \mathbb{R}^{n \times h}$ and degree parameter $p$, we compute $\tilde{\mathbf{Q}} \in \mathbb{R}^{n \times r}$ and $\tilde{\mathbf{K}} \in \mathbb{R}^{n \times r}$ using the fast sketches presented in Ahle et al. (2020) so that $\tilde{\mathbf{Q}}\tilde{\mathbf{K}}^\mathsf{T} \approx \mathbf{Q}^{\otimes p}(\mathbf{K}^{\otimes p})^\mathsf{T}$. Importantly, we show that using $r$ much less than $h^p$, we can obtain a good approximation to the matrix $\mathbf{Q}^{\otimes p}(\mathbf{K}^{\otimes p})^\mathsf{T}$. We call the parameter $r$ as the *sketch size* and the sketch size required is only a function of the *multiplicative accuracy* parameter $\varepsilon$. Replacing the matrices $\mathbf{Q}^{\otimes p}$ and $\mathbf{K}^{\otimes p}$ with $\tilde{\mathbf{Q}}$ and $\tilde{\mathbf{K}}$, we get an algorithm that has a multiplicative dependence on the sketch size $r$ instead of $h^p$. However the sketch sizes we use in our implementations are not very good at preserving the dot products for vectors that have negative entries. Thus our implementations are essentially to be seen as an attention mechanism that is *inspired* by polynomial attention.

Another important issue pointed out by Hua et al. (2022) in the cumulative sum algorithm of Performer in the case of decoding-only transformers is the RNN style sequential dependence, which makes the training extremely slow. We show that a block-based linear time algorithm can be used instead of the cumulative sum algorithm employed in Performer and that this leads to a signifi-

cant speedups. Our block-based lower triangular multiplication algorithm also recontextualizes the chunk-based attention mechanism of works such as of Hua et al. (2022) as essentially applying a causal mask to the simple linear transformer and additionally using a quadratic local attention.

## 2 SKETCHES FOR POLYNOMIAL KERNEL

We saw that given $\mathbf{Q}, \mathbf{K}, \mathbf{V} \in \mathbb{R}^{n \times h}$, output of the polynomial attention mechanism can be *exactly* computed using $O(nh^{p+1})$ floating point operations. While the time complexity is linear in the context length, the algorithm is very slow even for modest head sizes such as $h = 64$ or $128$ when $p = 4$. Thus, we resort to approximating the polynomial attention matrix using sketches for the polynomial kernel, which we formally describe ahead. We first state the definition of a sketch that has the "Approximate Matrix Multiplication (AMM)" guarantee.

**Definition 2.1** (Approximate Matrix Multiplication)**.** Given parameters $n$, $h$ and $p$, a randomized sketching matrix $\mathbf{S} \in \mathbb{R}^{h^p \times r}$ has the $(\varepsilon, p)$-AMM property if given any two $n \times h$ matrices $\mathbf{A}$ and $\mathbf{B}$, with probability $\geq 9/10$ over the randomized sketching matrix $\mathbf{S}$, we have that

$$\|(\mathbf{A}^{\otimes p}\mathbf{S})(\mathbf{B}^{\otimes p}\mathbf{S})^{\mathsf{T}} - \mathbf{A}^{\otimes p}(\mathbf{B}^{\otimes p})^{\mathsf{T}}\|_{\mathsf{F}} \leq \varepsilon \|\mathbf{A}^{\otimes p}\|_{\mathsf{F}}\|\mathbf{B}^{\otimes p}\|_{\mathsf{F}}.$$

Here given a matrix $\mathbf{A}$, $\|\mathbf{A}\|_{\mathsf{F}}$ denotes the Frobenius norm defined as $(\sum_{i,j} \mathbf{A}_{i,j}^2)^{1/2}$. The number of columns $r$ is referred to as the *sketch size*.

In general, a sketch need not be limited to being a *linear map* of the $h^p$ dimensional vectors. Two important properties of a sketching distribution are (i) the sketch size $r$ as a function of the accuracy parameter $\varepsilon$ and (ii) the time required to compute $\mathbf{A}^{\otimes p}\mathbf{S}$ given an arbitrary matrix $\mathbf{A}$. Ideally, we want the matrix $\mathbf{S}$ to have a structure such that $\mathbf{A}^{\otimes p}\mathbf{S}$ can be computed without realizing the large matrix $\mathbf{A}^{\otimes p}$. Ahle et al. (2020) give constructions of different sketches that have both the properties that the sketch size $r$ is small and the matrix $\mathbf{A}^{\otimes p}\mathbf{S}$ can be computed quickly. We describe the main properties of one of their sketches below and and explain how to compute $\mathbf{A}^{\otimes p}\mathbf{S}$ given a matrix $\mathbf{A}$.

**Theorem 2.2.** *Given degree $p$ and an accuracy parameter $\varepsilon$, there is a sketch $\mathbf{S}$ with $r \leq Cp/\varepsilon^2$ columns such that $\mathbf{S}$ satisfies the $(\varepsilon, p)$-AMM property as defined above. Given an arbitrary matrix $\mathbf{A} \in \mathbb{R}^{n \times h}$, the matrix $\mathbf{A}^{\otimes p}\mathbf{S}$ can be computed using $O(pnm \log m + pnh \log h)$ operations.*

We obtain the theorem by instantiating the construction in Ahle et al. (2020) with the base sketch being the Subsampled Randomized Hadamard Transform (SRHT) and the internal sketches being TensorSRHT. We will now explain how the sketch computation works for $p = 2$ and how it is extended to general values of $p$ that are powers of 2. First we define SRHT and TensorSRHT which are used in the construction of the sketch $\mathbf{S}$.

**Definition 2.3** (SRHT)**.** Given a dimension parameter $h$ that is a power of 2 and sketch size $r$, the $h \times r$ SRHT matrix is defined as $\sqrt{1/r} \cdot (\mathbf{DHP})$ where $\mathbf{D}$ is an $h \times h$ diagonal matrix with independent $\pm 1$ random variables, $\mathbf{H}$ is an $h \times h$ Walsh-Hadamard matrix and $\mathbf{P}$ is an $h \times r$ matrix where each of the $r$ columns is independently sampled from the uniform distribution over the coordinate vectors.

**Definition 2.4** (TensorSRHT)**.** Given a dimension parameter $h$ that is a power of 2 and sketch size $r$, the TensorSRHT sketch is defined to be the tuple $(\mathbf{D}_1, \mathbf{D}_2, \mathbf{P}_1, \mathbf{P}_2)$, where $\mathbf{D}_1, \mathbf{D}_2$ are $h \times h$ diagonal matrices where each diagonal entry is an independent $\pm 1$ random variable and $\mathbf{P}_1, \mathbf{P}_2$ are $h \times r$ matrices with each of the columns is independently sampled from the uniform distribution over the coordinate vectors in $\mathbb{R}^h$. Given matrices $\mathbf{A}_1, \mathbf{A}_2 \in \mathbb{R}^{n \times h}$, the result of applying the TensorSRHT defined by the tuple $(\mathbf{D}_1, \mathbf{D}_2, \mathbf{P}_1, \mathbf{P}_2)$ is the $n \times h$ matrix

$$\sqrt{1/r}\left[(\mathbf{A}_1\mathbf{D}_1\mathbf{HP}_1) * (\mathbf{A}_2\mathbf{D}_2\mathbf{HP}_2)\right]$$

where $*$ denotes the entry-wise product (Hadamard product) of two matrices and $\mathbf{H}$ is the $h \times h$ Walsh-Hadamard matrix.

Let $\text{SRHT}_1$ and $\text{SRHT}_2$ denote two *independently* sampled SRHT sketches and $\text{TensorSRHT}_1$ be a TensorSRHT sketch sampled independently from the SRHTs. Then the polynomial sketch for degree 2 is defined as $\text{TensorSRHT}_1(\text{SRHT}_1(\mathbf{A}), \text{SRHT}_2(\mathbf{A}))$. We describe the procedure and extension to $p = 4$ in Figure 3. The construction extends to all $p$ that are powers of 2 in a similar way.

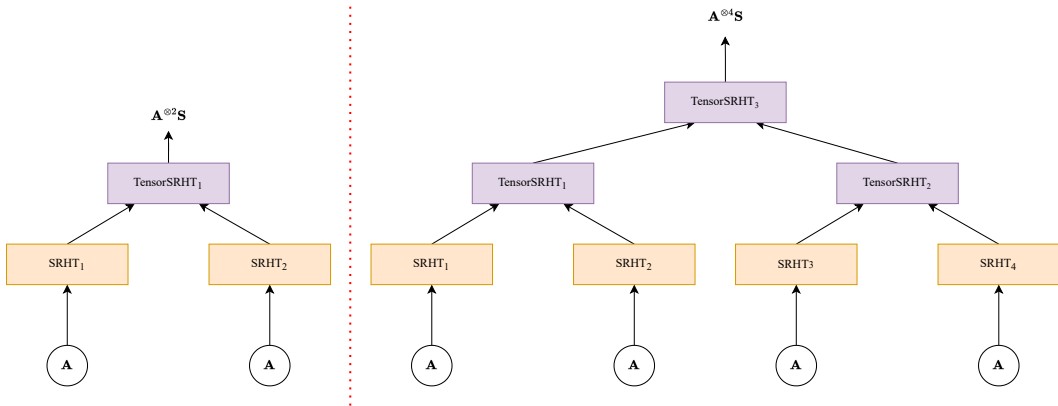

Figure 3: Sketches for polynomials of degrees $p = 2$ and $p = 4$.

## 2.1 NON-NEGATIVE ATTENTION APPROXIMATIONS

Given matrices $n \times h$ matrices $\mathbf{Q}$ and $\mathbf{K}$, the polynomial sketch described above can be used to approximate $(\mathbf{Q}\mathbf{K}^\mathsf{T})^p = \mathbf{Q}^{\otimes p}(\mathbf{K}^{\otimes p})^\mathsf{T}$ with $(\mathbf{Q}^{\otimes p}\mathbf{S})(\mathbf{K}^{\otimes p}\mathbf{S})^\mathsf{T}$. One issue with the polynomial sketches is that they do not preserve nonnegativity: while for $p$ even, the entries of the matrix $(\mathbf{Q}\mathbf{K}^\mathsf{T})^p$ are nonnegative, the entries of the matrix $(\mathbf{Q}^{\otimes p}\mathbf{S})(\mathbf{K}^{\otimes p}\mathbf{S})^\mathsf{T}$ need not be nonnegative.

The rows of the attention-weights matrix have the natural interpretation of each defining a probability distribution. If we instead use the sketch-based approximations, the approximate attention-weights matrix need not have nonnegative entries and hence present issues in the optimization of the models. Choromanski et al. (2020) solve this issue by using a feature map that maps to only nonnegative coordinates. Since the dot product of any two nonnegative vectors is nonnegative, their approximate attention matrix has only nonnegative coordinates.

But it is not necessary for the vectors to have only nonnegative coordinates for the dot products to be nonnegative. Consider arbitrary vectors $\mathbf{q}, \mathbf{k}$. The dot product $\langle \mathbf{q}^{\otimes 2}, \mathbf{k}^{\otimes 2} \rangle = \langle \mathbf{q}, \mathbf{k} \rangle^2 \geq 0$. Thus, given matrices $\mathbf{Q}^{\otimes p}\mathbf{S}$ and $\mathbf{K}^{\otimes p}\mathbf{S}$, consider the matrix $(\mathbf{Q}^{\otimes p}\mathbf{S})^{\otimes 2}((\mathbf{K}^{\otimes p}\mathbf{S})^{\otimes 2})^\mathsf{T}$. Since all the entries of the matrix are of the form $\langle \mathbf{q}^{\otimes 2}, \mathbf{k}^{\otimes 2} \rangle$ for some vectors $\mathbf{q}, \mathbf{k}$, all the entries of the matrix $(\mathbf{Q}^{\otimes p}\mathbf{S})^{\otimes 2}((\mathbf{K}^{\otimes p}\mathbf{S})^{\otimes 2})^\mathsf{T}$ are nonnegative as well. The "tensoring" trick ensures that all the entries in the approximate attention matrix are nonnegative at the cost of *squaring* the sketch size.

We now prove Theorem 2.5 which shows that a degree $p$ polynomial sketch followed by "tensoring" gives a degree $2p$ polynomial sketch.

**Theorem 2.5.** *Let $\mathbf{S} \in h^p \times r$ be an arbitrary oblivious sketch that satisfies the $(\varepsilon, \delta, t)$-JL moment and $(\varepsilon, \delta, 2t)$-JL moment properties for some even integer $t$. Given arbitrary matrices $\mathbf{C}$ and $\mathbf{D}$ with $h^p$ columns, we have that with probability $\geq 1 - \delta$,*

$$\|(\mathbf{C}\mathbf{S})^{\otimes 2}((\mathbf{D}\mathbf{S})^{\otimes 2})^\mathsf{T} - \mathbf{C}^{\otimes 2}(\mathbf{D}^{\otimes 2})^\mathsf{T}\|_\mathsf{F} \leq \sqrt{5}\varepsilon\|\mathbf{C}^{\otimes 2}\|_\mathsf{F}\|\mathbf{D}^{\otimes 2}\|_\mathsf{F}.$$

Results from Section 4 of Ahle et al. (2020) can be used to show that degree-$p$ polynomial sketch as constructed in Figure 3 with sketch size $r = \Omega(\varepsilon^{-2}p\log(1/\varepsilon\delta))$ satisfies the requirements of the above theorem thus giving a sketch that gives a nonnegative approximate attention matrix which provably approximates the degree $2p$ attention matrix.

## 3 A FAST ALGORITHM FOR LOWER TRIANGULAR (LT) MULTIPLICATION

Given *query* and *key* matrices $\mathbf{Q}$ and $\mathbf{K}$, using the sketches described in previous section, we can obtain matrices $\tilde{\mathbf{Q}}$ and $\tilde{\mathbf{K}}$ such that $(\mathbf{Q}\mathbf{K}^\mathsf{T})^{2p} \approx \tilde{\mathbf{Q}}\tilde{\mathbf{K}}^\mathsf{T}$ by computing sketches for degree $p$ polynomial and tensoring the sketches with themselves. To compute the output of the approximate attention mechanism in the causal setting (1), we need to compute the matrix $\mathsf{lt}_\triangle(\tilde{\mathbf{Q}}\tilde{\mathbf{K}}^\mathsf{T})\mathbf{V}$. Naively computing this matrix requires computing the $n \times n$ matrix $\tilde{\mathbf{Q}}\tilde{\mathbf{K}}^\mathsf{T}$, which is prohibitive. We now describe the cumulative sum algorithm as used in Choromanski et al. (2020) that can compute $\mathsf{lt}_\triangle(\tilde{\mathbf{Q}}\tilde{\mathbf{K}}^\mathsf{T})\mathbf{V}$ without computing the $n \times n$ matrix $\tilde{\mathbf{Q}}\tilde{\mathbf{K}}^\mathsf{T}$.

Given arbitrary matrices $\mathbf{A}, \mathbf{B} \in \mathbb{R}^{n \times r}$ and $\mathbf{C} \in \mathbb{R}^{n \times d}$, the $n \times d$ matrix $\mathsf{lt}_\triangle(\mathbf{A}\mathbf{B}^\mathsf{T})\mathbf{C}$ can be computed as follows: For $i = 1, \ldots, n$, define $\mathbf{D}_i = \sum_{j=1}^{i}(\mathbf{B}_{j,:})^\mathsf{T}\mathbf{C}_{j,:}$. Then the $i$-th row of the

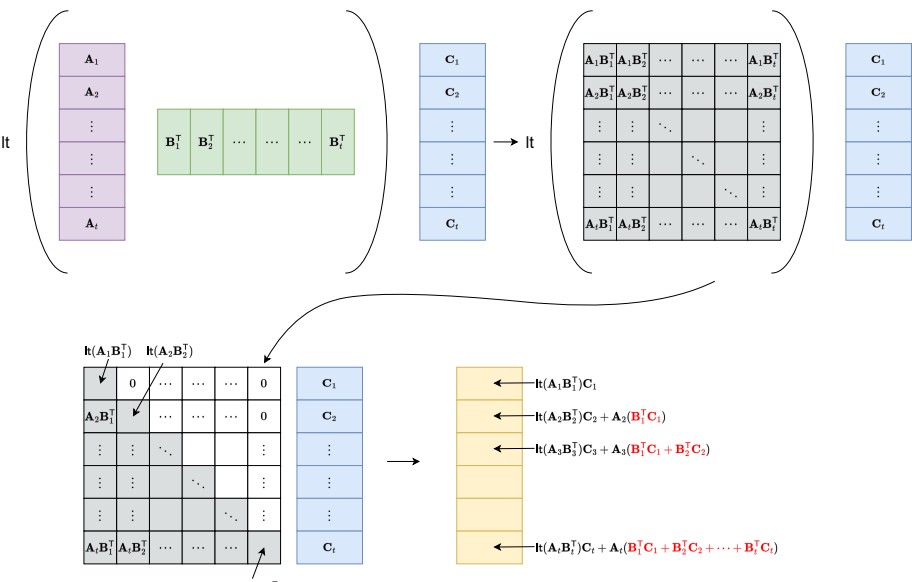

Figure 4: Block wise Lower Triangular Multiplication. Each of the matrices $\mathbf{A}_i$, $\mathbf{B}_i$ and $\mathbf{C}_i$ are blocks of the matrices $\mathbf{A}$, $\mathbf{B}$ and $\mathbf{C}$ respectively and each block has $b = n/t$ rows.

matrix $\mathsf{lt}_{\triangle}(\mathbf{AB}^{\mathsf{T}})\mathbf{C}$ is given by $\mathbf{A}_{i,:}\mathbf{D}_i$. We can first compute the matrices $(\mathbf{B}_{j,:})^{\mathsf{T}}\mathbf{C}_{j,:}$ using $O(nrd)$ operations and we can then compute the matrices $\mathbf{D}_1, \ldots, \mathbf{D}_n$ using another $O(nrd)$ operations using a simple cumulative sum algorithm. Overall, the matrix $\mathsf{lt}_{\triangle}(\mathbf{AB}^{\mathsf{T}})\mathbf{C}$ can be computed using a total of $O(nrd)$ operations. Choromanski et al. (2020) use this algorithm in the forward pass and show that a similar algorithm can be used to compute gradients in the backward pass.

This algorithm is quite slow on ML accelerators such as TPUs/GPUs because it reads/writes $n$ matrices of size $r \times d$ from/to the HBM. We propose using a block-based algorithm to reduce the number of read/write operations by a significant factor thereby improving the training latency by a large factor. Our algorithm is described in Figure 4. As we increase the block size used, we decrease the number of sequentially dependent steps. Now, it is clear that the following result holds.

**Theorem 3.1.** *Given matrices $\mathbf{A}$, $\mathbf{B} \in \mathbb{R}^{n \times r}$ and $\mathbf{C} \in \mathbb{R}^{n \times d}$ and a block size parameter $b$ which divides $n$, we can compute $\mathsf{lt}_{\triangle}(\mathbf{AB}^{\mathsf{T}})\mathbf{C}$ using $O(nr(b + d))$ operations.*

While setting $b$ to be 1 decreases the number of operations to be performed, the algorithm is slower on GPUs/TPUs because of the RNN style sequential dependence that is introduced by the cumulative sum. In our experiments, we set $b$ as 256 similar to the chunk size used by Hua et al. (2022).

## 4    POLYSKETCHFORMER

Using the polynomial sketch we described in Section 2 and the block-based algorithm to implicitly apply causal attention mask described in Section 3 we define a new fast attention mechanism for decoder-only transformers. We call the attention mechanism *polysketch* and the transformer augmented with polysketch attention as *PolySketchFormer*.

Consider a multi-layer Transformer model with each layer having multiple attention heads. In PolySketchFormer, for each attention head in each layer, we independently sample a sketching matrix $\mathbf{S}$ for polynomial kernel as described in Section 2. We now describe the output of the polysketch attention mechanism given the inputs $\mathbf{Q}, \mathbf{K}, \mathbf{V} \in \mathbb{R}^{n \times h}$. We compute matrices $\tilde{\mathbf{Q}} := (\mathbf{Q}^{\otimes p}\mathbf{S})^{\otimes 2}$ and $\tilde{\mathbf{K}} := (\mathbf{K}^{\otimes p}\mathbf{S})^{\otimes 2}$. Now define $\hat{\mathbf{V}} = [\mathbf{V}\ \mathbf{1}]$. Using the block-based lower triangular multiplication algorithm from Section 3, we compute the matrix $\mathsf{lt}_{\triangle}(\tilde{\mathbf{Q}}\tilde{\mathbf{K}}^{\mathsf{T}})\hat{\mathbf{V}} = [\mathsf{lt}_{\triangle}(\tilde{\mathbf{Q}}\tilde{\mathbf{K}}^{\mathsf{T}})\mathbf{V}\ \mathsf{lt}_{\triangle}(\tilde{\mathbf{Q}}\tilde{\mathbf{K}}^{\mathsf{T}})\mathbf{1}]$ and then the output of the polysketch attention mechanism is defined as $\mathbf{O} := \mathrm{diag}(\mathsf{lt}_{\triangle}(\tilde{\mathbf{Q}}\tilde{\mathbf{K}}^{\mathsf{T}})\mathbf{1})^{-1} \left[ \mathsf{lt}_{\triangle}(\tilde{\mathbf{Q}}\tilde{\mathbf{K}}^{\mathsf{T}})\mathbf{V} \right]$. By the definition of the matrices $\tilde{\mathbf{Q}}, \tilde{\mathbf{K}}$, we can see that

$$\mathbf{O}_{i,:} = \frac{\sum_{j \leq i} \langle \mathbf{Q}_{i,:}^{\otimes p}\mathbf{S}, \mathbf{K}_{j,:}^{\otimes p}\mathbf{S} \rangle^2 \mathbf{V}_{j,:}}{\sum_{j \leq i} \langle \mathbf{Q}_{i,:}^{\otimes p}\mathbf{S}, \mathbf{K}_{j,:}^{\otimes p}\mathbf{S} \rangle^2}. \tag{2}$$

We note that the above expression is similar to the output of the softmax attention mechanism with the major change being the replacement of the term $\exp(\langle \mathbf{Q}_{i,:}, \mathbf{K}_{j,:} \rangle / \sqrt{h})$ with $\langle \mathbf{Q}_{i,:}^{\otimes p}\mathbf{S}, \mathbf{K}_{j,:}^{\otimes p}\mathbf{S} \rangle^2$.

We note some important details of our implementation of polysketch attention:

**Degree of the polynomial.** In all our experiments, we use polynomial of degree 4 since Figure 2 suggests a degree 4 polynomial attention mechanism has perplexities comparable to that of the softmax attention. Accordingly, for the polysketch attention mechanism, we use Polynomial Sketches from Ahle et al. (2020) for degree 2 polynomials and by *tensoring* the sketches to obtain matrices that can approximate the polynomial attention with degree 4.

**Sketch Size.** In all our experiments, we use the sketches from Ahle et al. (2020) for degree 2 polynomial with a sketch size of 32. Since we *square* the sketch by tensoring the rows, we obtain matrices $\tilde{\mathbf{Q}}$ and $\tilde{\mathbf{K}}$ that have 1024 columns each. Further increasing sketch size to larger values leads to a slow-down in the algorithm since the time complexity of the lower triangular multiplication algorithm depends on the number of columns in the matrices $\tilde{\mathbf{Q}}$ and $\tilde{\mathbf{K}}$.

**Block Size in LT multiplication.** In all our experiments, we implement the lower triangular matrix multiplication algorithm of Section 3 with block size parameter $b$ as 256. As mentioned earlier, while using a small block size minimizes the number of floating point operations to be performed, it increases the sequential dependence and the number of reads/writes from/to the HBM. Thus we do not want to use a small value of $b$. On the flip side, a large block size $b$ requires a larger number of floating point operations to be performed and in the limit of $b$ being $n$, the lower triangular multiplication requires $O(n^2)$ time which is what we want to avoid. An additional important parameter in determining the block size that is to be used is the size of the smallest matrices that can be *efficiently* multiplied on the ML accelerator being used.

**Rematerialization.** We do not store the intermediate values that arise in the polysketch attention computation procedure for the backward pass. Instead we rerun the attention computation during the backward pass and compute the gradients. This lets us considerably decrease the amount of memory required to train the models with a small loss in performance.

## 5 EXPERIMENTS

**Datasets.** We train all our language models separately on the Wiki-40B (Guo et al., 2020) and PG-19 (Rae et al., 2019) datasets. We tokenize the datasets using SentencePiece vocabularies of size 32,000.

**Model Description, Experiment Setup and Implementation.** We train Transformer models that have 12 layers that each have 12 attention heads. We use an embedding dimension of 768 so that the head size is 768/12 = 64. We add sinusoidal position embeddings to the token embeddings at the start of the model and use Rotary Position Embeddings (Su et al., 2021) as the relative position embedding for each attention head in the model. We use Gated Linear Units (Dauphin et al., 2017; Shazeer, 2020) with an expansion factor of 4 as the FeedForward layer in the network. This configuration has 110M parameters. All models are trained for 125k steps using Adam optimizer with weight decay and a peak learning rate of 7e-4. We use 10k warmup steps and a linear learning rate schedule.

We perform extensive experiments on this model using different attention mechanisms such as softmax, polynomial, Performer, FlashAttention, and our Polysketch attention and at various context lengths. Finally, we run some experiments on models with more/fewer layers to test how polysketch scales to deeper layers and even longer contexts. We stress that the only difference between the models is the attention mechanism.

Our implementation of PolySketchFormer is written in JAX. In our experiments, we use a Pallas implementation (JAX authors, 2023) of FlashAttention and a JAX implementation of Performer open-sourced by the authors (Choromanski et al., 2020). All the experiments are conducted on 32 A100 GPUs with 40GB VRAM each.

**Results.** On the main model configuration of 12 layers, 12 attention heads per layer and an embedding dimension of 768, we report the perplexities of the models in Table 1 and training steps per second in Table 2. We note that we use a batch size of 8 per device for context length 512, 4 per device for 1024, 2 per device for 2048 and 1 per device for context lengths 4096, 8192 and 16384.

We observe that in all the cases across both the datasets, degree-4 polynomial attention achieves perplexities that are close to that of the softmax attention and that perplexities achieved by the models using Polysketch attention are close to the perplexities attained by softmax Transformer. Addition-

| | Wiki-40B | | | | | | PG-19 | | | | | |
|---|---|---|---|---|---|---|---|---|---|---|---|---|
| | 512 | 1k | 2k | 4k | 8k | 16k | 512 | 1k | 2k | 4k | 8k | 16k |
| Softmax | 16.5 | 16.0 | 15.6 | 15.2 | - | - | 15.5 | 14.5 | 14.5 | 12.8 | - | - |
| Polynomial | 16.6 | 16.2 | 15.6 | 15.4 | - | - | 15.6 | 14.9 | 14.6 | 13.1 | - | - |
| Polysketch | 18.2 | 17.6 | 17.2 | 17.6 | 18.2 | 17.7 | 16.5 | 17.6 | 16.2 | 14.8 | 15.4 | 15.8 |

Table 1: Perplexities on the eval splits of models separately trained on Wiki-40B and PG-19 datasets at various context lengths. Softmax and Polynomial models go out of memory when the context lengths are 8k and 16k. Using checkpointing to save memory for softmax and polynomial models, the training latency becomes too high to obtain results in a reasonable amount of time.

ally, we note the general trend[5] of better perplexities for models trained with longer contexts for all the attention mechanisms. We do not report perplexities for Performer since we observe that there is a data leak in the open-sourced version of Performer which inadvertently leads to information flow from future tokens during training which violates the causal masking necessary for auto-regressive language modeling. We note that in all our experiments, throughout the training, the perplexities of Polysketch attention models on eval split remains within 2-3 points of softmax Transformer.

The training latency results from Table 2 show that Polysketch attention is slower than FlashAttention for all context lengths up to $4k$ and that Polysketch attention becomes significantly faster than FlashAttention for context lengths $8k$ and beyond. We do note that our implementation of Polysketch is not optimized to the extent of FlashAttention and it maybe possible to get better performance with Polysketch at even smaller context lengths.

**Longer Contexts/Larger Models.** We train a 4-layer model, with otherwise same parameters as above models, with a context length 32k to observe the steps/sec achieved by PolySketchFormer. We observe that PolySketchFormer trains at 1.54 steps/sec and Transformer with FlashAttention trains at 0.34 steps/sec i.e., a 4.5x-speedup in this case.

On the other hand, we train a larger model (730M parameters), with 24 layers and an embedding dimension 1536, on Wiki-40B dataset to observe if deep PolySketchFormers track the performance of deep softmax Transformers. We observe that softmax Transformer has a perplexity of 14.4 on eval split and PolySketchFormer has a perplexity of 14.6 which provides evidence that PolySketchFormer can be used for training large models.

| Context Length | Softmax | Polynomial | Polysketch | Performer | FlashAttention |
|---|---|---|---|---|---|
| 512 | 6.25 | 6.2 | 3.7 | 2.3 | 5.8 |
| 1024 | 4.5 | 4.5 | 3.4 | 2.4 | 5.3 |
| 2048 | 4.3 | 4.1 | 3.3 | 1.8 | 4.7 |
| 4096 | 3 | 3.2 | 3.1 | 1.2 | 3.5 |
| 8192 | OOM | OOM | 2 | 0.7 | 1.4 |
| 16384 | OOM | OOM | 1.1 | 0.35 | 0.44 |

Table 2: Train Steps per Second on 32 A100 GPUs. We again stress that the batch size per device is kept same across the 8k and 16k experiments which leads to smaller number of steps per second.

## 6 CONCLUSIONS AND DIRECTIONS FOR FUTURE WORK

As our experiments show, *PolySketchFormer* offers significant gains in training latency for large context transformers. We note that a more careful implementation of our algorithm akin to FlashAttention can present even more gains than what our experiments suggest. Our perplexity results suggest that the performance of PolySketchFormer may not be too far away from the vanilla transformer architecture with softmax. We note that there are a lot of avenues to be explored to bring the perplexities of *PolySketchFormer* closer to that of the vanilla softmax Transformer. Some directions include (i) experimenting with higher degree polynomial sketches, (ii) training the network initially for a few iterations using quadratic-time polynomial attention and then train using the linear-time polysketch attention for rest of the iterations, etc. We leave these directions for future work.

---

[5]Although with some exceptions in the case of PolySketchFormer

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

# A  PROOF OF THEOREM 2.5

First, we define the $(\varepsilon, \delta, t)$-JL moment property. In the following, given a scalar random variable $\mathbf{X}$ and $t \geq 1$, $\|\mathbf{X}\|_{L^t}$ is defined to be $\mathbf{E}[|\mathbf{X}|^t]^{1/t}$. $\|\cdot\|_{L^t}$ defines a norm over random variables defined over the same sample space and in particular satisfies $\|\mathbf{X} + \mathbf{Y}\|_{L^t} \leq \|\mathbf{X}\|_{L^t} + \|\mathbf{Y}\|_{L^t}$.

**Definition A.1.** (JL-moment property) Given $\varepsilon, \delta \geq 0$ and $t \geq 1$, a random matrix $\mathbf{S}^{m \times r}$ has the $(\varepsilon, \delta, t)$-JL moment property if for any $x \in \mathbb{R}^m$ with $\|x\|_2$,

$$\|\|\mathbf{x}^\mathsf{T}\mathbf{S}\|_2^2 - 1\|_{L^t} \leq \varepsilon\delta^{1/t}.$$

We first note the following fact: If $\mathbf{S}$ has $(\varepsilon, \delta, t)$-JL moment property, then for any two arbitrary vectors $\mathbf{x}$ and $\mathbf{y}$, we have that $\|\langle \mathbf{S}^\mathsf{T}\mathbf{x}, \mathbf{S}^\mathsf{T}\mathbf{y}\rangle - \langle \mathbf{x}, \mathbf{y}\rangle\|_{L^t} \leq \varepsilon\delta^{1/t}\|\mathbf{x}\|_2\|\mathbf{y}\|_2$. For a proof see Lemma 9 from Ahle et al. (2020).

*Proof of Theorem 2.5.* Let $\mathbf{c}_i$ denote the $i$-th row of $\mathbf{C}$ and $\mathbf{d}_j$ denote the $j$-th row of $\mathbf{D}$. Then the $(i, j)$-th entry of the matrix $\mathbf{C}^{\otimes 2}(\mathbf{D}^{\otimes 2})^\mathsf{T}$ is equal to $\langle \mathbf{c}_i, \mathbf{d}_j\rangle^2$. Similarly, the $(i, j)$-th coordinate of the matrix $(\mathbf{CS})^{\otimes 2}((\mathbf{DS})^{\otimes 2})^\mathsf{T}$ is equal to $\langle \mathbf{S}^\mathsf{T}\mathbf{c}_i, \mathbf{S}^\mathsf{T}\mathbf{d}_j\rangle^2$ and therefore

$$\|(\mathbf{CS})^{\otimes 2}((\mathbf{DS})^{\otimes 2})^\mathsf{T} - \mathbf{C}^{\otimes 2}(\mathbf{D}^{\otimes 2})^\mathsf{T}\|_\mathsf{F}^2 = \sum_{i,j}(\langle \mathbf{S}^\mathsf{T}\mathbf{c}_i, \mathbf{S}^\mathsf{T}\mathbf{d}_j\rangle^2 - \langle \mathbf{c}_i, \mathbf{d}_j\rangle^2)^2.$$

Recall that given an integer $t \geq 1$, for a random variable $\mathbf{X}$, we define $\|\mathbf{X}\|_{L^t}$ as $\mathbf{E}[|\mathbf{X}|^t]^{1/t}$. Also note that $\|\mathbf{X}\|_{L^t}$ is a norm over the random variables and in-particular satisfies the triangle inequality. Now,

$$\|\|(\mathbf{CS})^{\otimes 2}((\mathbf{DS})^{\otimes 2})^\mathsf{T} - \mathbf{C}^{\otimes 2}(\mathbf{D}^{\otimes 2})^\mathsf{T}\|_\mathsf{F}\|_{L^t} = \|\|(\mathbf{CS})^{\otimes 2}((\mathbf{DS})^{\otimes 2})^\mathsf{T} - \mathbf{C}^{\otimes 2}(\mathbf{D}^{\otimes 2})^\mathsf{T}\|_\mathsf{F}^2\|_{L^{t/2}}^{1/2}$$

$$= \|\sum_{i,j}(\langle \mathbf{S}^\mathsf{T}\mathbf{c}_i, \mathbf{S}^\mathsf{T}\mathbf{d}_j\rangle^2 - \langle \mathbf{c}_i, \mathbf{d}_j\rangle^2)^2\|_{L^{t/2}}^{1/2}$$

$$\leq (\sum_{i,j}\|(\langle \mathbf{S}^\mathsf{T}\mathbf{c}_i, \mathbf{S}^\mathsf{T}\mathbf{d}_j\rangle^2 - \langle \mathbf{c}_i, \mathbf{d}_j\rangle^2)^2\|_{L^{t/2}})^{1/2}$$

where we used the triangle inequality of $\|\cdot\|_{L^t}$ in the last inequality. Now consider a single term $\|(\langle \mathbf{S}^\mathsf{T}\mathbf{c}_i, \mathbf{S}^\mathsf{T}\mathbf{d}_j\rangle^2 - \langle \mathbf{c}_i, \mathbf{d}_j\rangle^2)^2\|_{L^{t/2}}$. First, we have

$$(\langle \mathbf{S}^\mathsf{T}\mathbf{c}_i, \mathbf{S}^\mathsf{T}\mathbf{d}_j\rangle^2 - \langle \mathbf{c}_i, \mathbf{d}_j\rangle^2)^2$$
$$= (\langle \mathbf{S}^\mathsf{T}\mathbf{c}_i, \mathbf{S}^\mathsf{T}\mathbf{d}_j\rangle + \langle \mathbf{c}_i, \mathbf{d}_j\rangle)^2(\langle \mathbf{S}^\mathsf{T}\mathbf{c}_i, \mathbf{S}^\mathsf{T}\mathbf{d}_j\rangle - \langle \mathbf{c}_i, \mathbf{d}_j\rangle)^2$$
$$= (\langle \mathbf{S}^\mathsf{T}\mathbf{c}_i, \mathbf{S}^\mathsf{T}\mathbf{d}_j\rangle - \langle \mathbf{c}_i, \mathbf{d}_j\rangle + 2\langle \mathbf{c}_i, \mathbf{d}_j\rangle)^2(\langle \mathbf{S}^\mathsf{T}\mathbf{c}_i, \mathbf{S}^\mathsf{T}\mathbf{d}_j\rangle - \langle \mathbf{c}_i, \mathbf{d}_j\rangle)^2$$
$$\leq (1 + C)(\langle \mathbf{S}^\mathsf{T}\mathbf{c}_i, \mathbf{S}^\mathsf{T}\mathbf{d}_j\rangle - \langle \mathbf{c}_i, \mathbf{d}_j\rangle)^4 + 4(1 + 1/C)\langle \mathbf{c}_i, \mathbf{d}_j\rangle^2(\langle \mathbf{S}^\mathsf{T}\mathbf{c}_i, \mathbf{S}^\mathsf{T}\mathbf{d}_j\rangle - \langle \mathbf{c}_i, \mathbf{d}_j\rangle)^2$$

with probability 1 for any $C \geq 1$. Since both LHS and RHS are *non-negative* random variables, we obtain that

$$\|(\langle \mathbf{S}^\mathsf{T}\mathbf{c}_i, \mathbf{S}^\mathsf{T}\mathbf{d}_j\rangle^2 - \langle \mathbf{c}_i, \mathbf{d}_j\rangle^2)^2\|_{L^{t/2}}$$
$$\leq (1 + C)\|(\langle \mathbf{S}^\mathsf{T}\mathbf{c}_i, \mathbf{S}^\mathsf{T}\mathbf{d}_j\rangle - \langle \mathbf{c}_i, \mathbf{d}_j\rangle)^4\|_{L^{t/2}} + 4(1 + 1/C)\langle \mathbf{c}_i, \mathbf{d}_j\rangle^2\|(\langle \mathbf{S}^\mathsf{T}\mathbf{c}_i, \mathbf{S}^\mathsf{T}\mathbf{d}_j\rangle - \langle \mathbf{c}_i, \mathbf{d}_j\rangle)^2\|_{L^{t/2}}.$$

Now,

$$\|(\langle \mathbf{S}^\mathsf{T}\mathbf{c}_i, \mathbf{S}^\mathsf{T}\mathbf{d}_j\rangle - \langle \mathbf{c}_i, \mathbf{d}_j\rangle)^4\|_{L^{t/2}} = \|\langle \mathbf{S}^\mathsf{T}\mathbf{c}_i, \mathbf{S}^\mathsf{T}\mathbf{d}_j\rangle - \langle \mathbf{c}_i, \mathbf{d}_j\rangle\|_{L^{2t}}^4$$
$$\leq \varepsilon^4\delta^{2/t}\|\mathbf{c}_i\|_2^4\|\mathbf{d}_j\|_2^4$$

assuming that $S$ has $(\varepsilon, \delta, 2t)$-JL moment property. We also have

$$\|(\langle \mathbf{S}^\mathsf{T}\mathbf{c}_i, \mathbf{S}^\mathsf{T}\mathbf{d}_j\rangle - \langle \mathbf{c}_i, \mathbf{d}_j\rangle)^2\|_{L^{t/2}} = \|\langle \mathbf{S}^\mathsf{T}\mathbf{c}_i, \mathbf{S}^\mathsf{T}\mathbf{d}_j\rangle - \langle \mathbf{c}_i, \mathbf{d}_j\rangle\|_{L^t}^2$$
$$\leq \varepsilon^2\delta^{2/t}\|c_i\|_2^2\|d_j\|_2^2$$

assuming that $\mathbf{S}$ has $(\varepsilon, \delta, t)$-JL moment property. Overall, we get

$$\|(\langle \mathbf{S}^\mathsf{T}\mathbf{c}_i, \mathbf{S}^\mathsf{T}\mathbf{d}_j\rangle^2 - \langle \mathbf{c}_i, \mathbf{d}_j\rangle^2)^2\|_{L^{t/2}}$$
$$\leq (1 + C)\varepsilon^4\delta^{2/t}\|\mathbf{c}_i\|_2^4\|\mathbf{d}_j\|_2^4 + 4(1 + 1/C)\langle \mathbf{c}_i, \mathbf{d}_j\rangle^2\varepsilon^2\delta^{2/t}\|\mathbf{c}_i\|_2^2\|\mathbf{d}_j\|_2^2.$$

Picking $C = 1/\varepsilon$ and assuming $\varepsilon \leq 1/5$, we get that

$$\|(\langle \mathbf{S}^\mathsf{T} \mathbf{c}_i \rangle^2 - \langle \mathbf{c}_i, \mathbf{d}_j \rangle^2)^2\|_{L^{t/2}} \leq 5\varepsilon^2 \delta^{2/t} \|\mathbf{c}_i\|_2^4 \|\mathbf{d}_j\|_2^4.$$

Thus, we have

$$\|\|(\mathbf{CS})^{\otimes 2}((\mathbf{DS})^{\otimes 2})^\mathsf{T} - \mathbf{C}^{\otimes 2}(\mathbf{D}^{\otimes 2})^\mathsf{T}\|_\mathsf{F}\|_{L^t} \leq \sqrt{5}\varepsilon\delta^{1/t} \sqrt{\sum_{i,j} \|\mathbf{c}_i\|_2^4 \|\mathbf{d}_j\|_2^4}$$

$$\leq \sqrt{5}\varepsilon\delta^{1/t} \|\mathbf{C}^{\otimes 2}\|_\mathsf{F} \|\mathbf{D}^{\otimes 2}\|_\mathsf{F}.$$

By using Markov's inequality, we obtain that with probability $\geq 1 - \delta$,

$$\|(\mathbf{CS})^{\otimes 2}((\mathbf{DS})^{\otimes 2})^\mathsf{T} - \mathbf{C}^{\otimes 2}(\mathbf{D}^{\otimes 2})^\mathsf{T}\|_\mathsf{F} \leq \sqrt{5}\varepsilon \|\mathbf{C}^{\otimes 2}\|_\mathsf{F} \|\mathbf{D}^{\otimes 2}\|_\mathsf{F}. \qquad \square$$

## B   PROOF OF THEOREM 3.1

*Proof.* Let $t = n/b$ be the number of blocks. We see that we perform $t$ matrix multiplications of the form $[b, r, b]$ (which denotes a multiplication of matrices of shapes $b \times r$ and $r \times b$), $t$ matrix multiplications of the form $[r, b, d]$, a cumulative sum of $t - 1$ matrices of shape $[r, d]$, $t - 1$ matrix multiplications of the form $[b, r, d]$ and finally $t - 1$ matrix additions of the shape $[b, d]$. Overall the number of operations performed is $O(tb^2 r + tbrd + trd + tbrd + tbd) = O(nbr + nrd)$. $\qquad \square$

If $\mathbf{S}$ has the so-called strong $(\varepsilon/e, \delta^2)$-JL moment property (Ahle et al., 2020, Definition 19), then it follows that $\mathbf{S}$ has both $(\varepsilon, \delta, \log(1/\delta))$-JL moment and $(\varepsilon, \delta, 2\log(1/\delta))$-JL moment properties. Results from Section 4 of Ahle et al. (2020) can be used to show that degree-$p$ polynomial sketch as constructed in Figure 3 with sketch size $r = \Omega(\varepsilon^{-2} p \log(1/\varepsilon\delta))$ satisfies the requirements of this theorem.

