# OpenReview forum: "PolySketchFormer: Fast Transformers via Sketches for Polynomial Kernels"
_ICLR.cc/2024/Conference — Submitted to ICLR 2024_

### Official Review · Reviewer_vEUf · 2023-10-31

**Soundness:** 3 good
**Presentation:** 3 good
**Contribution:** 4 excellent
**Rating:** 8
**Confidence:** 4

**Summary:**

Transformers have an inherent quadratic complexity in context length due to self-attention computation with softmax function. Softmax self-attention can be approximated using polynomial functions. This method has complexity linear in context length, but the vanilla application of polynomial approximation creates an exponential dependency on the degree $p$ of the polynomial.
This paper introduces a sketch-based approximation for polynomial kernel self-attention using the *oblivious sketching* methods for higher-order polynomial kernels described in *Ahle et al. (2020)*. This method can bypass the exponential dependency on degree $p$ while the approximation error of the polynomial kernel self-attention matrix is inversely correlated with sketch size. They validate their theoretical guarantees by comparing the perplexity metric of softmax, polynomial kernel, and the new algorithm.
Additionally, the authors provide a block-based algorithm for decoder-only transformer models. This circumvents the sequential nature of the cumulative-sum algorithm used in prior approaches such as *Performer*.
Experiments show that combining these two methods shows decent speed-up gains in runtime compared to standard softmax self-attention, polynomial kernel self-attention as well as fast algorithms for self-attention such as Performer and Flash Attention.

**Strengths:**

**Originality**

The clever combination of oblivious sketching for polynomial kernels with polynomial kernel self-attention although both of these individual ideas have been known for quite a while shows the originality of this paper.

**Quality**

Solid theoretical results on approximation guarantees for polynomial kernel self-attention and the experiments demonstrate the gains obtained by algorithms.

**Significance**

Quadratic complexity in self-attention is a significant barrier in scaling transformer-based models in low-resource settings. Softmax self-attention has shown superior empirical accuracy over other types of self-attention, but approximating softmax self-attention is known to be hard in subquadratic time in context length. Polysketch self-attention performs comparably to softmax while using truly subquadratic time and I believe this is a significant contribution.

**Weaknesses:**

Presentation and clarity of polysketch self-attention can be improved.

Typically in matrix sketching, we take the product of $A^{\otimes p} \in \mathbb{R}^{n \times h^p}$ and $S \in \mathbb{R}^{h^p \times r}$ explicitly. However, the proposed algorithm does not perform this explicit multiplication and avoids the exponential runtime dependency on $p$.  I believe the algorithm described in *Ahle et al. (2020)* uses the Fast Fourier Transform to compute $A^{\otimes p} S$ efficiently with the tree-based algorithm. However, this paper does not have any indication of how this product is computed. I believe it is important to outline this process for the completeness of this paper.
I am surprised that the authors do not emphasize this fact enough, since it is a significant component that enables the removal of exponential dependency on degree $p$.

Typos:
1. On page 12, proof of theorem 2.5, Definition A.1. "$x \in \mathbb{R}^m$ with $\lvert\lvert x \rvert\rvert_2$" $\rightarrow$ "$x \in \mathbb{R}^m$ with $\lvert\lvert x \rvert\rvert_2 = 1$"
2. On page 13, the paragraph after the proof of theorem 3.1 does not make sense. I believe it belongs earlier in the paper or is redundant.

**Questions:**

**Questions**

1. In theorem 2.2, how large can the constant $C$ be? If $C$ is large, $r$ may become too large for it to be practical.
2. In theorem 2.2, what is $m$?
3. Perhaps I am missing something but on page 12, in the proof of theorem 2.5, does the JL-moment property of $S$ imply that $\langle S^Tc_i, S^Td_j \rangle.\langle c_i, d_j \rangle \le \langle c_i, d_j \rangle^2$ ? I do not see how $(\langle S^Tc_i, S^Td_j \rangle - \langle c_i, d_j \rangle + 2 \langle c_i, d_j \rangle )^2 (\langle S^Tc_i, S^Td_j \rangle - \langle c_i, d_j \rangle)^2 \le (1+C) (\langle S^Tc_i, S^Td_j \rangle - \langle c_i, d_j \rangle)^4 + 4(1+1/C) \langle c_i, d_j \rangle^2 (\langle S^Tc_i, S^Td_j \rangle - \langle c_i, d_j \rangle)^2$ holds for any $C \ge 1$ otherwise.

**Additional remarks**

Perhaps the following remarks can be candidates for future work.

1. Even though the block-based method is targeted at decoder-only models, I believe the approximate algorithm for polynomial kernel self-attention can be applied in general to any transformer-based architecture. Have the authors explored any other architectures with this approximate self-attention algorithm?
2. In "On The Computational Complexity of Self-Attention [Feyza Duman Keles, Pruthuvi Mahesakya Wijewardena, Chinmay Hegde]" paper, a polynomial approximation to softmax self-attention is derived using the finite Taylor series which includes lower order terms as well. Technically, this should be able to get arbitrarily close to softmax self-attention as the order $p$ is increased. If we use Polysketch self-attention to approximate each term of the Taylor series approximation, and thereby approximate softmax self-attention, I am curious what would be the closest approximation one can get while still maintaining subquadratic complexity in context length and subexponential complexity in degree $p$.

---

> ### Author Response · Authors · 2023-11-16
>
> - **“does not have any indication of how this product is computed”**: We use the algorithm as exactly described in the paper of Ahle et al., which avoids explicit computation of the large matrix to apply the sketch. We try to explain this algorithm in Figure 3 (Definitions of SRHT and TensorSRHT are presented in Definition 2.3 and 2.4). We will include a more detailed description of how exactly it is to be implemented to avoid the exponential dependence in p.
>
> - **Theorem 2.2**: Our apologies for the typo. It should be r instead of m.
>
> - **Proof of Theorem 2.5**:  We use AM-GM inequality to obtain the following inequality for any C > 0: for any $a, b \ge 0$, $(a + b)^2 = a^2 + b^2 + 2ab = a^2 + b^2 + 2(a/\sqrt{C})(b\cdot\sqrt{C}) \le a^2 + b^2 + (a^2/ C) + (b^2 C) \le (1 + 1/C)a^2 + (1 + C)b^2$
>
> We use this to obtain that $(\langle S^T c_i, S^T d_j \rangle - \langle c_i, d_j\rangle + 2\langle c_i, d_j\rangle)^2 \le (1 + C)(\langle S^T c_i, S^T d_j\rangle - \langle c_i, d_j\rangle)^2 + (1 + 1/C)(2\langle c_i, d_j\rangle)^2$ from which the inequality follows. So we do not need the assumption that $\langle S^Tc_i, S^Td_j\rangle \cdot \langle c_i, d_j\rangle \le \langle c_i, d_j\rangle^2$.
>
> - **Other Self-Attention Mechanisms**: Yes, our sketch-based attention mechanism can definitely be used outside the causal language modeling that we focus on in this paper. We plan to explore this in future.
>
> - **Closest Approximation to Softmax using Polynomial**: It would definitely be interesting to study how close we can get to softmax using these techniques. But the lower bounds in the work of Alman and Song suggest that we shouldn’t be able to get too close to Softmax attention output in sub-quadratic time unless the entries of the query and key vectors are bounded.

---

> > ### Comment · Reviewer_vEUf · 2023-11-22
> >
> > I thank the authors for the clarification on the proof. Please add more details on how exactly the sketch is computed in the next version.

---

### Official Review · Reviewer_YWkN · 2023-11-01

**Soundness:** 3 good
**Presentation:** 4 excellent
**Contribution:** 3 good
**Rating:** 5
**Confidence:** 3

**Summary:**

The authors study the issue of quadratic complexity for transformers. The authors propose the use of polynomial kernels and sketching to replace the softmax function to achieve linear complexity. For the theoretical part, they show that there are guarantees on the approximation error of the sketch. For the empirical part, they show the speed of the proposed method is faster than the compared methods for longer contexts.

**Strengths:**

1. The proposed method seems to be simple and reasonable.
2. The authors provide guarantees on the approximation of sketching.

**Weaknesses:**

1. The empirical result is weak. It seems that the proposed method with 8k, 16k context is slower than the other method with 2k, 4k context and has higher perplexity. Doesn’t this mean the proposed method is not better?
2. Need to elaborate more on the data leak of Performer, so the reviewers can verify the statement.
3. The authors need to elaborate on why some newer methods that perform better on the long range arena does not need to be compared. For example, ChordMixer or LongFormer.
4. The speedup does not seem to be large. Thus, it feels important to have more experiments to verify the perplexity/perform drop is generally within the acceptable range.

**Questions:**

1. Can you elaborate more on the data leak of Performer? Is this your discovery or it has been pointed out by others? It is better to give the reviewers some ways to validate this statement. Some reference/link to the issue will help.
2. Is rematerialization also used in the other methods? It is a bit unclear from the writing.
3. Can you briefly explain why Performer is slower than the softmax baseline in the experiments? This does not seem to be the case in the original paper. Is this because of bad implementation?

---

> ### Author Response · Authors · 2023-11-16
>
> - **Empirical Results**: The table does suggest that our longer context models have worse perplexity. But we note that we have not tried to pick the best possible parameters for each context length and we trained at all context lengths using the same hyperparameters (learning rate schedule, peak learning rate, number of steps, etc.) which can be the reason for the trends.
>
> In some situations such as when working with long documents, one might want to work with models trained on long contexts even if they have worse perplexities than models trained with shorter contexts since we cannot fit the whole document otherwise. We do also note that for longer contexts, our model trains significantly faster than vanilla transformer/Flash Attention so our models can be trained for significantly more steps using the same amount of compute time. Thus, training for a larger number of steps, we may be able to beat softmax transformers though we have not yet fully explored this direction yet.
>
> - **Data Leak in Performer**: This is an issue discovered by us since there was a mismatch between the eval perplexities we computed vs the quality of text generated by the model trained using performer attention. We will include a full explanation of the bug in the next version of the paper. Here we include a brief description of the issue.
>
> Consider a two token context length. Let $q_1$ and $q_2$ be the queries and $k_1$ and $k_2$ be the keys and $v_1$ and $v_2$ be the values. Performer first maps the queries and keys to $q’_1$ and $q’_2$ and $k’_1$ and $k’_2$ respectively using random projections. For the keys, they finally need to compute $\exp(-\|k_1\|_2^2) * \exp(k_1’)$ and $\exp(-\|k_2\|_2^2) * \exp(k_2’)$ where $\exp(k’)$ means applying the exponential function entrywise to coordinates of $k’$. So as to not create large entries and make the training stable they compute the maximum value of an entry in $k_1’$ and $k_2’$ and subtract the same value from all the entries in $k_1’$ and $k_2’$ essentially making the largest entry to be 0 and so when we apply exponential function, we will not get large values. Mathematically, subtracting the same quantity should not matter because after applying exponential, it is essentially just multiplying all the entries with the same quantity and hence the attention matrix they compute should not change. But observe that the entries in $k_1’$ may get subtracted from the largest entry in $k_2’$ and since we are working with limited precision, this creates a channel using which information from future keys leak to present attention calculation.
>
> We observe that the model tries to exploit this channel and therefore gets better eval perplexities since it is using information from future tokens at test time to predict the same tokens. We reported this bug to the authors of Performer paper and they confirmed the bug. We do note that the bug is not present in the non-causal version of the Performer. But since our paper focuses on the causal version of the performer, we are unable to report the numbers for the correct version of the performer implementation. We did implement Performer attention removing the bug and using an alternate “normalization” to stabilize training but the training latency was significantly worse and we couldn’t finish training in a reasonable amount of time to be able to compare with our models. We note that the running times that we reported are for the buggy version of Performer which is publicly available.
>
> - **Rematerialization**: Rematerialization is used in Our model to recompute the output of the polysketch attention mechanism. Recomputation is also used in the Performer implementation and FlashAttention implementation.
>
> - **Long Range Arena**: The benchmarks in Long Range Arena are for the non causal attention setting. Since we focused on improving causal language modeling (autoregressive decoding) in this work, we do not evaluate our mechanism in tasks in the Long Range Arena benchmark.
>
> - **Speedup is Not Large**: We note that we do obtain significant speedups and the speedups improve as we increase the context lengths. For example, we report that on a 4 layer model with a context length 32k, we obtain a 4.5x reduction in training latency compared to FlashAttention and on the main model we study (similar parameters to GPT2-small), we get a 2x reduction. Thus, we can train our models with twice as many iterations in the same amount of time as we can train a Flash Attention model.
>
> - **Performer being Slow**: The runtimes we reported are directly using the Publicly available version of Performer (which as we note has a data leak.). Depending on the model parameters, the cumulative sum algorithm used in Performer can be significantly slow because of the large number of reads/writes it has to perform to compute the output of the attention mechanism. This was also observed by the authors of “FLASH : Transformer Quality in Linear Time”.

---

> > ### Comment · Reviewer_YWkN · 2023-11-22
> >
> > I thank the authors for the response. The clarification and the endeavors of the authors convince me to keep me original score.
> > Otherwise, I would considering lowering it, as the longer context models of the authors have worse perplexity than the others with shorter contexts is a significant weakness to me.
> >
> > I understand the difficulty to do hyperparameter tuning under insufficient computing resources. I also agree longer context models might have its use case even when having worse perplexities. However, if sacrifice in performance is allowed, I believe there will be many more straightforward ways to scale up to longer context that can be compared to. Thus, I believe the authors have to show some cases (even simpler ones) where they do have improvement over other shorter context ones in order to make the paper more convincing.

---

### Official Review · Reviewer_B6vB · 2023-11-01

**Soundness:** 3 good
**Presentation:** 3 good
**Contribution:** 3 good
**Rating:** 6
**Confidence:** 3

**Summary:**

This paper proposes a new transformer architecture called PolySketchFormer that uses polynomial sketching techniques to approximate the softmax attention mechanism efficiently.  To be concrete, the authors approximate the dot products among tensorized queries and keys efficiently by first lowering their dimensions through randomized sketching matrices. Then, the authors ensure non-negativity of the approximated attention scores by a  "tensoring trick". Additionally, the authors propose a block-based algorithm for multiplication with lower-triangular causal masks, which is faster, though using more operations, than Performer (Choromanski et al., 2020).  Experiments are conducted to verify PolySketchFormer's effectiveness, speed, and ability to handle long-range context.

**Strengths:**

- S1. Using randomized sketch to reduce the  $O(nh^{p+1})$  complexity is insightful.
- S2. The proposed procedure for computing $\mathrm{It}(\tilde{Q}\tilde{K}^{T})V$ is contributive, especially to other other linear Transformers as well.
- S3. PolySketchFormer's effectiveness, speed, and ability to handle long-range context is verified by experiments.

**Weaknesses:**

- W1. According to Definition 2.4,  PolySketchFormer can only express/approximate the polynomial $f(s)=s^p$ ($s:= \langle q,k \rangle$), where $p$ is **a power of 2**. This limits the flexibility to "approximate Softmax".
- W2. The output and input of the SRHT and TensorSRHT can be better illustrated. It is a bit confusing at first glance.
- W3. A typo: On Page 3, the second to last line, it should be $q\in \mathbb{R}^h$, right?
- W4. In experiments, PolySketchFormer is compared only with Performer and FlashAttention. More efficient Transformers should be involved.

**Questions:**

- Q1. Please check W1.
- Q2. In Table 1, what is exactly the Softmax model? Is it FlashAttention?

---

> ### Author Response · Authors · 2023-11-16
>
> - **"Limited flexibility to approximate Softmax"**: We do not think of polynomial attention as trying to approximate softmax but instead think of it as an entirely different attention mechanism. Softmax attention puts more emphasis on large dot products and similarly a high degree polynomial attention will also put more emphasis on large dot products though the main difference is that polynomial attention also amplifies dot products with large negative values. Our hypothesis is that the query and key vectors obtained when training with polynomial attention don’t create large negative dot products and hence may not present large issues.
>
> - **Typo**: Yes, it should be $q \in \mathbb{R}^h$
>
> - **“In Table 1, what is exactly the Softmax model? Is it FlashAttention?”**: The softmax models are trained with a default implementation of dot product attention in Flax. This implementation does not have all the optimizations performed by FlashAttention. A properly implemented FlashAttention model should also achieve the same perplexity values.

---

### Official Review · Reviewer_Jyhm · 2023-11-01

**Soundness:** 2 fair
**Presentation:** 2 fair
**Contribution:** 3 good
**Rating:** 3
**Confidence:** 3

**Summary:**

This paper introduces a new architecture that attempts to be close to the original attention, but without the quadratic computational cost. The first observation relies on the cost of computing the attention matrix and performing the softmax on this matrix. The idea in this work is to replace the softmax with a polynomial of degree-k (where k is even). The second observation lies on approximating the polynomial computation with sketches. Then, to preserve the nonegativity, a squaring step after the computation of the first sketch is used. Lastly, various small architecture-optimization steps are completed to ensure that the implementation is fast, including the step of fast lower triangular computation of the causal attention matrix or the block-based variant inspired by FlashAttention.

**Strengths:**

The quadratic cost of the transformer is indeed a major bottleneck in the literature recently, and this is further evident in the context lengths used in practical models. In that sense, reducing this cost to linear is a relevant topic in the literature. In addition, I have not noticed any similar approach in the literature so far using the sketches to approximate the polynomial that approximates the attention matrix in turn.

**Weaknesses:**

- Even though the paper proposes a way to avoid the nonnegativity, another property of the softmax is that it upper-bounds a value to at most 1, while it’s less clear how the proposed model deals with this or whether this is even an issue.

- The paper mentions in the introduction that in practice most sota papers use the vanilla transformer in practice (i.e. large-scale experiments), but then there are no sota papers and large-scale experiments conducted in the experimental section. In addition, there are few to no ablations on the choices made in this work.

- There are some related works in the direction of polynomial and efficient attention that might be worth comparing with: “Hydra Attention: Efficient Attention with Many Heads” (10/2022) and “Linear Complexity Self-Attention with 3rd Order Polynomials” (3/2023).

- What is the performance of the proposed model in actual NLP-related tasks, such as translation etc? The paper mentions that this method is not only applicable on generation, but also conditional generation, but I do not see any related results here.

**Questions:**

- What is the perplexity of the model and why is this a good metric?

- Why is a degree 4 polynomial used?

- Why are all experiments up until the 16k context window only? What’s the core limitation for extending it further?

- Why is a single-degree k polynomial (even more that it is restricted to be even) approximating the softmax? Are there any guarantees or insights on that?

---

> ### Author Response · Authors · 2023-11-16
>
> -  **"Upperbounds to 1"**: We normalize the attention matrix outputs to have a value 1 in our models as well. Concretely, given Q and K, we use sketches + tensoring to get Q’ and K’ and then to compute the final output we run our algorithm to get LT(Q’ K’^T)*[V 1] i.e., we append a column of 1s to the V matrix and the last column of the result gives the necessary values we need to use to normalize the rows of the output. So, the approximate attention matrix we implicitly apply to V to get the output also has the sum of each row as 1.
>
> -  **Comparison with SOTA models on NLP tasks**: As the focus of our work was on improving the training latency for language models, we could not compare on benchmarks with SOTA models that are quite large. We note that we train the models from scratch on two datasets for only 125k iterations and hence cannot be competitive with SOTA models (even with similar model configuration as us as they are trained for much longer on more diverse datasets) on usual long context benchmarks which study for e.g., how good the model is at summarizing long documents, how good the model is at retrieving tokens, or translation etc.
>
> - **"HydraAttention and Related Mechanisms"**: Thanks for the references! We looked at HydraAttention and it appears that the paper focuses on improving Vision Transformers instead of language modeling and at the first glance it seems to us that such models may be a bit too weak for causal language modeling which is our main focus. Our intuition is that the attention heads for language modeling seem to require a somewhat large (like 64/128) number of features to get something useful from the attention mechanism. But we agree it is something we should explore more.
>
>
> - **"Fast Conditional Generation"**: Since the KV cache for linear transformers essentially reduces to storing a single matrix with size independent of context length as opposed to in the vanilla softmax models which need to store key, value pairs for all the tokens in the context to perform inference, the linear transformers are significantly less memory hungry during inference. The focus of our paper was to mainly improve training latency, so we did not include the improved memory requirements/inference latency for our models.
>
> - **Perplexity**: Perplexity is generally used as a measure of the quality of a language model. It is defined as the exponential of the average cross entropy loss of the model on the training dataset. One can roughly think of perplexity as the “inverse probability assigned by the model to the correct token as the next token” given the context. If a model has perplexity 100, then the model essentially only assigns on average (in the geometric mean sense) a probability of 1/100 to the correct token whereas a model that has perplexity 2, assigns on average a probability of ½ to the correct token.
>
> We agree a single measure, such as perplexity, is not good at predicting how good the model is but it is known to be a good proxy for the performance and is used as the main way to compare language models in earlier works on efficient mechanisms for training language models such as “FLASH : Transformer Quality in Linear Time” and references therein.
>
> - **Degree 4**: We performed experiments with degrees 2, 4 and 8 and found that degree 2 polynomial attention is not close in performance to softmax attention whereas degree 4 is very close as we report in the paper. We also observed that going to higher degrees does not help much since degree 4 is already very close to softmax. So we settle to use degree 4 polynomial in our models.
>
> - **Experiments only up to 16k**: Since we trained all our models using data parallelism, the context length is limited only by the 40GB VRAM available in the A100 GPUs we use. There is nothing stopping us from extending it to longer contexts using multiple GPUs and using pipeline parallelism and the same improvements in the single device setting should carry over. We note that we do report training latency results for training a smaller 4-layer model with a context length of 32k in the paper.
>
> - **"Why is a single-degree k polynomial approximating the softmax?"**: We note that we are not claiming that polynomial attention is approximating softmax attention. We think of polynomial attention as an entirely new attention mechanism which also works. Intuitively, the softmax is used to convert a vector of numbers into a probability distribution. Nothing in principle is stopping us from using other functions, such as polynomials in our case, from converting a vector of numbers into a probability distribution. Our other intuition is that softmax forces the probability distribution to focus on the vector entry with the largest value which is also done by a high degree polynomial as well although even-degree polynomials also amplify large negative entries. So the key, query dot products function somewhat differently under softmax and polynomial attentions.

---

> > ### Comment · Reviewer_Jyhm · 2023-11-21
> > **Response**
> >
> > Dear authors,
> >
> > I am thankful for the thoughtful responses. However, many of my questions are still not addressed.
> >
> > The responses note that the method trains faster for longer contexts, "so our models can be trained for significantly more steps using the same amount of compute time". However, the fact that it trains faster does not mean that it can generalize better.
> >
> > Also, as the reviewer YWkN points out, some more recent methods are not evaluated, which should be included. The responses point out that this can be attributed to the focus on causal masks, but this is simply a technique for training. SImilarly, I am not completely settled on the 16k context length.
> >
> > Beyond that, the responses mention that the point of this paper is simply to improve the "training latency, so we did not include the improved memory requirements/inference latency for our models". At the same time the model only trains the models "for only 125k iterations", thus we are not sure what a fair comparison with trained models is.
> >
> > The claim "degree 4 is already very close to softmax" is not very clear to me. What does it mean very close to softmax? In what sense and by which metrics?

---

> > > ### Author Response · Authors · 2023-11-21
> > > **Our Clarifications**
> > >
> > > Thanks a lot for going through our rebuttal! We will try to clarify your questions. Please ask us if you have further questions.
> > >
> > > - The overall idea of the paper is to introduce an alternative attention mechanism (polysketch) and show that it can be a suitable replacement for softmax attention, which is used in state of the art language models, to train a decoder-only transformer while being significantly faster to train at long context lengths in practice, and get around a complexity-theoretic barrier of using softmax for attention in theory. To show that this can be a suitable replacement in practice, we train small language models with both softmax attention and our polysketch attention mechanism, and show using perplexities on eval data as a proxy, that the polysketch attention mechanism can be an alternative for training language models. We agree that “*training faster*” does not mean it generalizes better and the results in Table-1 do suggest that when softmax models and polysketch models are trained for the *same number of iterations*, then the polysketch models lag behind the softmax models. By “so our models can be trained for significantly more steps using the same amount of compute time”, we meant that it may be possible to bridge this gap or even beat the  performance of softmax models, since we can train polysketch models for a lot more iterations as compared to softmax models, owing to significantly lower training latency. Another way the performance of softmax models can be beat is by training a larger model using polysketch attention, which is made possible by the improved training latency of our attention mechanism. For example, the 24-layer model we report in page 9, trained using polysketch attention mechanism has a better perplexity on the eval dataset as compared to any of the softmax transformer models in Table-1.
> > >
> > > - At inference time, softmax transformers need to store key and value vectors in a cache to speedup the prediction of next tokens, and the memory requirement for this task scales linearly in the context length. Whereas all kernel based linear transformers, including ours, need to store only a small matrix to predict next tokens given the context. This is a *secondary* benefit of our models i.e., our models are less memory hungry during inference in addition to the *primary* benefit of being able to be *trained faster*. Since the focus of this paper is on improving training latency, we do not report any experiments of benefits during inference. But based on the reasons explained above, we expect to see similar improvements in inference time, and will add them to the paper.
> > >
> > > - “Degree 4 is already very close to softmax”: Sorry if the statement was confusing. We report a comparison of softmax and polynomial (degree 4) models in Table-1. In terms of perplexities on eval datasets, the models trained with degree-4 polynomial attention are very close to the models trained with softmax attention on those data sets. This is what we wanted to convey in our rebuttal. We also trained models with degree-2 polynomial attention and degree-8 polynomial attention and observed that the models trained with degree-2 polynomial are worse, in terms of perplexities on eval datasets, and models trained with degree-8 polynomial are not a lot better, again in terms of perplexities on eval datasets. This convinced us that degree-4 polynomial models are the right thing to target to balance the tradeoff between generalization performance and training latency.
> > >
> > > - 16k context: Sorry, we do not understand your confusion. Our context lengths are only limited by the GPU memory that is available. In general, large models are trained using multiple GPUs using pipeline parallelism or tensor parallelism. For the experiments in this paper, we limit ourselves to training models that fit into a single GPU which limits the context length to 16k for the model that we train. Our attention mechanism runs in time linear in the context length, and we will see training latency improvements even when we train models using multiple devices on even larger context lengths.

---

> > > > ### Author Response · Authors · 2023-11-22
> > > > **Reminder**
> > > >
> > > > A reminder that the discussion period ends today. Please let us know if you have any questions. We performed some preliminary **inference** experiments with our models at a context length of 64k. We observed that the time to generate the first token given a prompt of length 64k for a polysketch model is 822ms whereas it is 2.24s for a FlashAttention model which shows our model is about 3x faster. We stress that these are only times to generate one token given the prompt and we need to do more extensive experiments to evaluate the time necessary to generate 'x' number of tokens since the key/value pairs are cached and the entire $n \times n$ attention matrix need not be recomputed for inference of each new token. We will add these experimental results to the paper.

---

### Official Review · Reviewer_q4q2 · 2023-11-02

**Soundness:** 4 excellent
**Presentation:** 3 good
**Contribution:** 2 fair
**Rating:** 3
**Confidence:** 4

**Summary:**

This paper proposes a new approach to attention computation in transformer networks that scales linearly rather than quadratically in the context length, and thus allows scaling to much larger context lengths.

The key idea is to replace the softmax non-linearity typically used in attention with a very simple polynomial non-linearity — in fact just f(x) = x^4. With this non-linearity, one can compute the attention matrix using a simple tensoring approach in O(n h^5) time, where n is the context length and h is the dimension of the token embeddings. This is linear in n, but the large dependence on h is bad in practice. To avoid this, the paper further sketches the tensored up query/key matrices using known techniques for sketching polynomial kernel matrices — in particular TensorSketch. This maintains linear in n scaling but improves the dependence on h to also linear, at the cost of some approximation and a quadratic dependence on a sketch size parameter that controls accuracy.

The paper also introduces a blocked variant of an existing ‘cumulative sum’ algorithm for applying a lower triangular to the approximate attention matrix. This blocked variant trades off parallelizability (i.e. efficiency on GPUS) for memory usage/runtime.

Overall, the idea of this paper is nice and simple, and it is well explained. None of the ideas are particularly novel — TensorSketch for example has been used frequently in the past to approximate matrices under polynomial/entrywise non-linearitiess, like in the attention setting. The blocked cumulative sum algorithm is a simple optimization of an existing approach.

The finding that replacing an exponential non-linearity with a much easier to compute/approximate polynomial is nice. Although I would have liked to see this discussed/analyzed more. Intuitively, f(x) = x^4 behaves very differently from f(x) = exp(x) since as x -> -infty, x^4 -> infty while exp(x) -> 0. That is, tokens with very different (opposite) embeddings in this new approach have large attention matrix values, rather than small ones. Why does this change not effect performance/cause issues with the new approach?

Given that the theoretical/algorithmic contributions are somewhat limited, I would have liked to see more of an empirical evaluation. This is beyond my area of expertise, however, the empirical evaluation seems to focus only on perplexity scores, which are a very limited metric, and as discussed in the paper itself, cannot necessarily distinguish methods that effectively implement long range attention from those that don’t. This seems to be a major draw back, especially given that the goal of the paper is to scale transformers to have longer context windows, and thus presumably longer range attention. From what I can tell there isn’t really any evidence in the empirical evaluation that the paper achieves/makes progress towards this goal.

The paper is fairly well written but has lots of typos and grammatical errors. It would need significant proofing before final publication.

**Strengths:**

See full review.

**Weaknesses:**

See full review.

**Questions:**

I have asked various questions/made specific comments below. I starred ones that are more important/would be good to see addressed in the author response period.

Questions/Comments:
- **In Fig. 1 I don’t understand how the vanilla softmax attention runs out of memory at context length of 4k. Shouldn’t the memory footprint just be roughly (4k)^2 doubles, or roughly 32 MBs? This is extremely small. I can compute a 4k x 4k attention matrix in a few seconds on my laptop.
- “because of an RNN-style sequential dependence of linear transformers which essentially renders the linear attention implementation to be memory-bandwidth bound” — I don’t understand what this means. Sequential dependence of what on what? Does this not depends on the algorithm used?
- The work of Han, Avron, Shin on approximating entrywise transformed lower rank matrices with tensor sketch seems very related but is not cited. See: https://proceedings.mlr.press/v119/han20a.html
- The parameter m in Theorem 2.2 (in the runtime) seems to be undefined. Is this meant to be r?
- **I didn’t follow the explanation at the bottom of page 5. If we break out exactly what this is doing it is computing:
[AD_1 H P_1 D_2 H P_2 * AD_3 H P_3 D_4 H P_4]
where D_1,P_1 and D_3,P_3 are the random matrices chosen for SRHT_1(A) and SRHT_2(A) respectively. And D_2,P_2 and D_4,P_4 are chosen by TensorSRHT_1. This doesn’t look right to me. What is the purpose of applying the ‘double sketch’ D_1 H P_1 D_2 H P_2. If one looks at the Ahl et al. paper (https://arxiv.org/pdf/1909.01410.pdf), the base sketch is applied already to the first level of tensor products, not to the original input. I.e. the base sketch sketches from h^2 to r dimensions. Not h to r dimensions. In particular, Def 15 of that paper defines TensorSHRT which matches Def 2.4 of this paper, and should be directly used for sketching A^{tensor 2}.
- In Theorem 2.5, the relevant JL moment properties are never defined or explained.
- I think as stated Theorem 3.1 is meaningless, as it doesn’t mention the memory tradeoff or even the algorithm used. So the theorem follows vacuously from noting that O(nr(b+d) >= O(nrd) and by Choromanski et al. we can comput mask(AB^T) C in O(nrb) time.
- ** In the end the tensor sketch is only actually applied for a degree 2 polynomial. Is it really the best option there? Why not explicitly compute Q^{tensor 2}, K^{tensor 2}. SVD them to get lower dimensional approximations tilde Q and tilde K and then compute tilde Q^{tensor 2} and tilde K^{tensor 2}. You are already explicitly tensoring up once in your last step anyways.
- Related to the above, what is the final runtime complexity of your algorithm? The naive tensoring method is O(n h^5). Your’s is what in terms of n, h, and the sketch size r? Roughly O(n(h+r^2))?

---

> ### Author Response · Authors · 2023-11-16
>
> - **Novelty**: We do agree that the use of tensorsketch and other polynomial sketches is not new. But we believe our new finding that the lack of “non-negativity” for the polynomial sketches is a hindrance to model convergence is an important point. And our simple fix to this using tensoring to ensure non-negativity is also we believe an important contribution and a useful technique in general.
>
> While the block-based algorithm to compute LT($AB^T$)$C$ is a simple modification of the existing algorithm, we want to make an important point that the slowness of kernel based linear transformers is not intrinsic and just changing the algorithm to ours can make it significantly faster in practice. The slowness of the cumulative sum algorithm was cited as the most important reason in the “FLASH : Transformer Quality in Linear Time” paper to explore other types of models but as we show we can significantly improve the running time just using a block based algorithm.
>
> - **Intuition for why polynomial works**: Yes, you are correct that the functions behave very differently as x goes to -infty. Our intuition is that the polynomial attention learns to make the key vectors, that it doesn’t want to put a probability mass on, close to orthogonal to the query vector and hence will assign a low probability under the polynomial distribution.
>
> - **Memory footprint of attention**: In order to train the models, we compute the gradient of the loss function with respect to the weights using standard backpropagation algorithm. The backpropagation algorithm needs to store all the results of the intermediate computation or recompute the activations from the checkpoints. In our model, with a context length of 4k, we have 12 layers each with 12 attention heads and hence the backpropagation algorithm needs to store 144 4k x 4k attention matrices and results of various other computations such as the key, query, value projections for each token in the context in each layer and the intermediate computations of the feedforward layers, etc. Our model with vanilla softmax attention runs out of memory to store all the required information for backpropagation at 4k context length.
>
> - **“RNN style Sequential Dependence”**: The cumulative sum algorithm used by earlier works to compute LT($AB^T$)C computes rows of the output matrix one-by-one and inorder to perform these operations, we have to read/write $n$ matrices to the high-bandwidth memory of the GPU where $n$ is the number of rows of $A$. This is what the authors of “FLASH : Transformer Quality In Linear Time'' refer to as “RNN style Sequential Dependence of Cumulative Sum algorithm on the context length” since RNNs also need to read and write $n$ matrices to the high-bandwidth memory where $n$ is the context length.
>
> 	As you say, the sequential dependence (as in the number of read/writes performed) does depend on the algorithm used to compute LT($AB^T$)$C$ and is not an inherent property of Linear Transformers.
>
> - **Missing Citations and Typos**: Thanks for suggesting the paper to us. We will add a citation in the next version. In theorem 2.2, yes the parameter should be r and not m.
>
> - **Double Sketch**: In the work of Ahle et al., they apply base sketch to obtain input sparsity time algorithms which is not very relevant in this work. As you observe, we can skip applying the base sketch and directly proceed to applying the TensorSRHT sketch.
>
> - **Theorem 3.1**: We wanted to convey that there is a block-based algorithm exclusively operating on blocks with b rows and computes LT($AB^T$)$C$ using those numbers of floating point operations and which needs $O(n/b)$ intermediate sequential reads/writes. We will update the theorem statement.
>
> - **Explicitly Computing Q^{tensor 2} and K^{tensor 2}**: In the model we use in our experiments, which is based on GPT-2 small, the $Q$ and $K$ matrices have 64 columns and in the current state of the art models (such as Palm-2), generally the $Q$, $K$ matrices have 128 columns. Explicitly tensoring these matrices would create matrices $Q’$ / $K’$ with 4096 (in 64 case) and 16384 (in 128 case) columns which is very slow to compute on GPUs and requires a large amount of memory as well. Our sketching technique will let us approximate attention even for larger head sizes such as 128 or more without such quadratic increase in the time/memory requirements.
>
> - **Final Complexity of PolySketch**: Yes, the final complexity of sketch + tensoring would be $O(n(h \log h) + nr^2)$ using a fast Hadamard product implementation.

---

> > ### Comment · Reviewer_q4q2 · 2023-11-16
> >
> > Thanks for the response which is helpful in clarifying some points.
> >
> > I didn't follow the comment on the base sketch: in your implementation do you apply it or not? I.e. was this just a typo in the described algorithm? Or is in initial sketch (which seems unneeded and in fact strictly worse than not-sketching in the base case) actually used in the implementation reported on?
> >
> > It would also be helpful to comment on the empirical evaluation in terms of perplexity scores.

---

> ### Author Response · Authors · 2023-11-16
>
> Thanks for going through the rebuttal!
>
> - **Base Sketch**: In our implementations for the paper, we used the base sketch. After submitting the paper, we realized that applying the base sketch was not necessary and can be removed. We want to note that including/removing the base sketch did not affect the quality of the models in our experiments.
>
> - **Empirical Evaluation**: Yes, perplexity as a single metric is not very useful in determining the quality of a model since models with small context lengths can also achieve good perplexities. In general, specific benchmarks such as LongBench (Bai et al., 2023), which includes tasks such as long document summarizing,  document question-answering, token recall capability, are used to measure how good large state-of-the-art models are at handling long contexts. In our case, since we train the models from scratch on a limited amount of data, we do not expect these small models to be good at handling benchmarks in LongBench. Our main aim was to show, using perplexities as a proxy for model quality as has been done in previous efficient attention mechanism papers such as "FLASH : Transformer Quality in Linear Time", that polysketch attention is a promising mechanism to scale transformers to long contexts, in the sense that the perplexity trends are similar to the softmax attention transformers while being significantly faster to train.

---

> > ### Author Response · Authors · 2023-11-22
> > **Reminder**
> >
> > A reminder that the discussion period ends today. Please let us know if you have any more questions. Thanks!

---

### Meta-Review · Area_Chair_GxRQ · 2023-12-06

**Metareview:**

The authors introduce PolySketchFormer, a transformer architecture that replaces the softmax in the transformer's attention mechanism with a degree-4 polynomial kernel for achieve $O(nh)$ forward pass complexity, where $n$ is the number of tokens and $h$ is the token dimension. While this technique has been visited numerous times in the literature, the main novelty here is twofold:
* A naive use of degree-4 polynomials results in a runtime that scales as $h^5$. The authors advocated the use of ``tensor sketches'' -- a known method in polynomial sketching -- to reduce this dependence to linear in $h$.
* The authors further propose a block-based algorithm for computing causal attention matrices, borrowing ideas from the architecural speedups used in implementing FlashAttention.

Many reviewers appreciated the direction taken by the authors. However, reviewers pointed out several drawbacks with the current version of the paper:
* The theory + algorithmic contributions were somewhat light, being (mostly) re-configurations of existing techniques.
* Therefore, the burden fell on the experimental component. Here, the paper fell quite short; the only comparisons were with Performer and with FlashAttention, ignoring the numerous other linear-time attention approaches already existing in the literature (some of which were pointed out in the reviews). The authors response (saying that comparisons with SOTA were outside scope) is insufficient in my opinion.
* The use of perplexity alone as an evaluation metric also raised concerns, with some reviewers calling for evaluations on downstream tasks.
The authors are advised to address these points for any future revisions.

**Justification For Why Not Higher Score:**

Overall sentiment was negative (despite the lengthy back and forth with the reviewers).

**Justification For Why Not Lower Score:**

N/A

---

### Decision · Program_Chairs · 2024-01-16

Reject